# Association between body mass index and atopic dermatitis among adolescents: Findings from a national cross-sectional study in Korea

**Jae Hyeok Lim[1,2], Yun Seo Jang[1,2], Dan Bi Kim[1,2], Suk-Yong Jang[2,3], Eun-Cheol Park[2,4] \***

**1** Department of Public Health, Graduate School, Yonsei University, Seoul, Republic of Korea, **2** Institute of Health Services Research, Yonsei University, Seoul, Republic of Korea, **3** Department of Healthcare Management, Graduate School of Public Health, Yonsei University, Seoul, Republic of Korea, **4** Department of Preventive Medicine, Yonsei University College of Medicine, Seoul, Republic of Korea

\* ECPARK@yuhs.ac

## Abstract

**Data Availability Statement:** The third-party data underlying the results presented in the study is publicly available from the KYRBS website (https://www.kdca.go.kr/yhs/home.jsp).

### Background

The association between atopic dermatitis and childhood overweight and obesity has been studied extensively, but the results are inconclusive; most studies have focused on body mass index as a measure of obesity, with few investigating the relationship with underweight. Therefore, this study aimed to investigate the association between body mass index levels and atopic dermatitis in Korean adolescents.

### Methods

3-year (2019–2021) of Korea Youth Risk Behavior Web-based Survey were used. Body mass index was used to measure obesity and a recent diagnosis within the past year was used as the criterion for atopic dermatitis. Multiple logistic regression analyses were performed to explore the associations. The odds ratios (ORs) and 95% confidence intervals (CIs) were calculated.

### Results

A total of 144,183 adolescents aged 12–18 years were included in this study (74,704 males and 69,479 females). Over the past year, 5.4% of males and 7.3% of females were diagnosed with atopic dermatitis in the study population. Adolescents with normal weight (males [OR: 1.19, CI: 1.02–1.38]; females [OR: 1.26, CI: 1.10–1.43]) and overweight (males [OR: 1.37, CI: 1.16–1.61]; females [OR: 1.37, CI: 1.19–1.58]) were more likely to develop atopic dermatitis than underweight.

**Funding:** The author(s) received no specific funding for this work.

**Competing interests:** The authors have declared that no competing interests exist.

## Conclusion

Increased degree of obesity may contribute to the development of atopic dermatitis. The normal-weight and obese adolescents had higher likelihood of developing atopic dermatitis compared with the underweight adolescents.

## Introduction

Atopic dermatitis (AD), also known as atopic eczema, is a common chronic inflammatory skin disease characterized by relapsing eczematous lesions and intense itching [1]. Internationally, the incidence rates of AD are approximately 15%–20% in children and 1%–3% in adults [2]. The Korean Disease Control and Prevention Agency (KDCA) conducted a survey on the prevalence of AD in adolescents, revealing a rate of 16.4% among individuals aged 12–18 years [3]. It typically develops in early infancy and often persists until adulthood [4]. Considering the "atopic march," which begins with AD and progresses to food allergy, allergic asthma, and allergic rhinitis in adulthood, it is crucial to administer treatment at the appropriate time in childhood [5]. Although AD is not a fatal disease, it causes a significant psychosocial burden, which can worsen the quality of life and mental health of patients and their caregivers (such as family and relatives) [6]. Moreover, AD imposes an economic burden, which involves direct costs for treatment and indirect costs such as absenteeism from work and school, leading to avoidance of social interactions [7].

The global prevalence of childhood obesity has substantially risen in recent decades [8], and it is linked to the emergence of various comorbidities [9]. Several studies have examined the association of AD with childhood overweight and obesity, but it is still unclear whether AD is one of these comorbidities. Some studies have reported no association between these factors [10, 11], while others have found a positive association [12–15]. Additionally, there are studies that have reported a positive association in only one sex [16–18]. In the meantime, the majority of those studies utilized body mass index (BMI) as the measure for obesity, with a limited number of studies examining the relationship with underweight. Among those studies, a large number of them did not define underweight, comparing the obese group with the non-obese group. Alternatively, the findings for underweight individuals were not statistically significant due to the limited sample size, or were not meaningfully interpreted, even though significant results were determined [19–21]. Given that BMI may not provide an accurate representation of body fat [12–14, 17], it is imperative to conduct an inquiry into the potential existence of risks within the realm of normal weight, which may present a threat to this correlation.

Furthermore, AD management aims to avoid or minimize events that can trigger itchiness and to provide continuous treatment of the disease. However, certain treatments for AD, such as phototherapy and pharmaceutical interventions, pose potential risks that may give rise to additional complications, prompting inquiries into their efficacy in managing the condition [22]. If the degree of obesity has a positive impact on AD, then weight reduction could be used as a non-pharmacological measure to support adolescent AD treatment on a broader scale. Therefore, the primary objective of this cross-sectional study was to investigate the association between the degree of obesity as measured by BMI and the morbidity of AD in a representative sample of Korean adolescents.

## Materials and methods

### Data

The study used data from the Korea Youth Risk Behavior Web-based Survey (KYRBS) conducted by the KDCA. The KYRBS is aimed at investigating the current risk behavior of Korean adolescents and identifying the health indicators in adolescents that can be used in health promotion program planning, assessment, and international comparison. The multistage stratified random cluster sampling technique was employed by the KYRBS to produce a sample that was nationally representative. To reduce sampling error, the sample was stratified by school type and geographic location (17 provinces). 400 middle schools and 400 high schools were selected annually for the sample allocation using proportional sampling in order to match the population. Previous studies demonstrated the validity and reliability of this survey [23, 24]. All participants answered the self-report online survey anonymously.

### Participants

This study utilized KYRBS data collected during a 3-year period (2019–2021). From the 3-year data, all 167,099 participants were initially considered eligible. After excluding adolescents who were missing in height and weight (N = 4,372) and adolescents who were missing in other variables (N = 18,589), a total of 144,138 participants were finally included in this study (50,445 in 2019, 45,351 in 2020, and 48,387 in 2021). Of the total participants, 74,704 were male adolescents, and 69,479 were female adolescents, with ages ranging from 12–18 years old. Ethics approval for the KYRBS was waived by the KDCA institutional review board (IRB) under the Bioethics & Safety Act and opened to the public for academic use. When participating in the KYRBS, all participants, as well as their parents or legal guardians, completed an informed consent.

### Variables

To measure the degree of obesity, we used the BMI, which is calculated by dividing an individual's weight (in kg) by their height squared (in meters). These were obtained by the response that adolescents who participated in the survey recorded their height and weight to the first decimal place. Based on the KDCA guidelines, the adolescent BMI was classified as underweight (under 5th percentile [pct]: BMI<5th), normal weight (5th–85th pct: 5th ≤BMI<85th), overweight (85th–95th pct: 85th ≤BMI<95th), and obesity (over 95th pct: 95th ≤BMI) [25]. In this study, the BMI levels were categorized as underweight (under 5th pct: BMI<5th), normal weight (5th–85th pct: 5th≤BMI<85th), and overweight (over 85th pct: 85th ≤BMI). For subgroup, BMI were categorized into six distinct groups: under 5th pct, 5th–25th pct, 25th–50th pct, 50th–75th pct, 75th–95th pct, and over 95th pct.

To determine the presence of physician-diagnosed AD as the dependent variable, we used two questions. The survey first inquired whether the participants had ever been diagnosed with AD. Second, it asked if they had received an AD diagnosis within the past year. Participants who answered "yes" to both questions were classified into the AD "yes" group. For those who responded "no" to the first question, the survey instructed them to skip the second question. The second question was only answered by those who responded "yes" to the first question. Eventually, those who did not answer "yes" to the second question were placed in the AD "no" group. For subgroups, AD diagnosis status were categorized into three distinct groups: never diagnosed, past diagnosed, and recent diagnosed.

The following factors based on previous studies were considered as covariates [26, 27]: sociodemographic factors (sex [male and female], grade [7th, 8th, 9th, 10th, 11th, and 12th], area

of residence [metropolitan, urban, or rural], household income level [high, middle, or low]). Health-related factors (secondhand smoke experience in a week [yes or no], frequency of physical activity [no exercise, 1–3 times a week, 4–6 times a week, or every day], sleeping time [$\geq$8 hours a day: sufficient or <8 hours a day: insufficient], subjective stress level (a lot, a little, or free), subjective health (healthy, normal, or unhealthy), frequency of soda/sweet beverage/fast-food intake (no, several times a week, or several times a day), lifetime diagnosis experience of asthma/rhinitis (yes or no), and year (2019, 2020, and 2021).

## Statistical analyses

All analyses were stratified by sex [16–18], since some studies reported sex differences in the incidence of AD. To explore the distribution of the adolescents' general characteristics, a chi-square test was used. To determine the association between BMI and AD, multiple logistic regression analysis was performed after adjusting for all covariates [28]. Moreover, multiple and multinomial logistic regression analyses of subgroups were conducted to evaluate the association between the stratified sections of AD diagnosis and BMI. The odds ratios (ORs) and 95% confidence intervals (CIs) were calculated to identify the associations among variables. SAS version 9.4M6 (SAS Institute, Cary, NC, USA) was used to perform all statistical analyses.

## Results

### Descriptive analyses

Table 1 shows a summary of the general characteristics of the study population by sex. Over the past year, 4,014 of the 74,704 male participants (5.4%) and 5,092 of the 69,479 female participants (7.3%) were diagnosed with AD. As the BMI levels increased (in the order of underweight, normal weight, and overweight), the proportion of patients diagnosed with AD increased in both male and female groups. The results of the chi-square test between BMI and AD in adolescents were significant (both p < .0001).

### Regression analyses

Table 2 shows the results of the multiple logistic regression analysis between BMI and AD. The male adolescents with normal weight (OR: 1.19, CI: 1.02–1.38) and overweight (OR: 1.37, CI: 1.16–1.61) had higher odds of developing AD compared with those who were underweight. Similarly, the female adolescents with normal weight (OR: 1.26, CI: 1.10–1.43) and overweight (OR: 1.37, CI: 1.19–1.58) also had higher odds of developing AD compared with those who were underweight.

### Subgroup analyses

Table 3 shows the significant results of subgroup analysis stratified by independent variables. In the health-related factors, adolescents who did not exercise had higher odds of developing AD compared with those who were underweight [male: normal weight (OR: 1.55 CI: 1.16–2.07) and had overweight (OR: 1.62, CI: 1.18–2.21); female: normal weight (OR: 1.23, CI: 1.02–1.47) overweight (OR: 1.45, CI: 1.19–1.77)]. Adolescents who had insufficient sleeping time had higher odds of developing AD compared with those who were underweight [male: normal weight (OR: 1.23, CI: 1.03–1.47) and had overweight (OR: 1.38, CI: 1.14–1.67); female: normal weight (OR: 1.29, CI: 1.12–1.49) and overweight (OR: 1.46, CI: 1.24–1.71)]. Adolescents with gender difference in subjective stress levels and subjective health perception showed variations in the odds of developing AD compared with the underweight groups. Subjective stress level: a lot [male: normal weight (OR: 1.55, CI: 1.18–2.03) and overweight (OR: 1.62, CI: 1.23–2.14)];

**Table 1. General characteristics of the study population.**

| Variables | Atopic dermatitis | | | | | | | | | | | | | |
|---|---|---|---|---|---|---|---|---|---|---|---|---|---|---|
| | Male | | | | | | | Female | | | | | | |
| | Total | | Yes | | No | | P-value | Total | | Yes | | No | | P-value |
| | N | % | N | % | N | % | | N | % | N | % | N | % | |
| Total (N = 144,138) | 74,704 | 100.0 | 4,014 | 5.4 | 70,690 | 94.6 | | 69,479 | 100.0 | 5,092 | 7.3 | 64,387 | 92.7 | |
| **Body Mass Index** | | | | | | | < .0001 | | | | | | | < .0001 |
| Underweight | 5,210 | 7.0 | 233 | 4.5 | 4,977 | 95.5 | | 5,547 | 8.0 | 334 | 6.0 | 5,213 | 94.0 | |
| Normal | 49,242 | 65.9 | 2,546 | 5.2 | 46,696 | 94.8 | | 51,759 | 74.5 | 3,755 | 7.3 | 48,004 | 92.7 | |
| Overweight | 20,252 | 27.1 | 1,235 | 6.1 | 19,017 | 93.9 | | 12,173 | 17.5 | 1,003 | 8.2 | 11,170 | 91.8 | |
| **Age** | | | | | | | 0.0533 | | | | | | | 0.0003 |
| 7th | 13,294 | 17.8 | 653 | 4.9 | 12,641 | 95.1 | | 12,571 | 18.1 | 817 | 6.5 | 11,754 | 93.5 | |
| 8th | 12,921 | 17.3 | 663 | 5.1 | 12,258 | 94.9 | | 12,250 | 17.6 | 861 | 7.0 | 11,389 | 93.0 | |
| 9th | 12,981 | 17.4 | 719 | 5.5 | 12,262 | 94.5 | | 11,844 | 17.0 | 868 | 7.3 | 10,976 | 92.7 | |
| 10th | 12,116 | 16.2 | 668 | 5.5 | 11,448 | 94.5 | | 11,007 | 15.8 | 851 | 7.7 | 10,156 | 92.3 | |
| 11th | 11,863 | 15.9 | 673 | 5.7 | 11,190 | 94.3 | | 11,020 | 15.9 | 857 | 7.8 | 10,163 | 92.2 | |
| 12th | 11,529 | 15.4 | 638 | 5.5 | 10,891 | 94.5 | | 10,787 | 15.5 | 838 | 7.8 | 9,949 | 92.2 | |
| **Region** | | | | | | | 0.4416 | | | | | | | 0.2636 |
| Metropolitan | 33,120 | 44.3 | 1,743 | 5.3 | 31,377 | 94.7 | | 30,065 | 43.3 | 2,151 | 7.2 | 27,914 | 92.8 | |
| Urban | 35,720 | 47.8 | 1,958 | 5.5 | 33,762 | 94.5 | | 34,099 | 49.1 | 2,554 | 7.5 | 31,545 | 92.5 | |
| Rural | 5,864 | 7.8 | 313 | 5.3 | 5,551 | 94.7 | | 5,315 | 7.6 | 387 | 7.3 | 4,928 | 92.7 | |
| **Household income** | | | | | | | < .0001 | | | | | | | < .0001 |
| High | 31,193 | 41.8 | 1,619 | 5.2 | 29,574 | 94.8 | | 25,580 | 36.8 | 1,839 | 7.2 | 23,741 | 92.8 | |
| Middle | 34,572 | 46.3 | 1,828 | 5.3 | 32,744 | 94.7 | | 35,528 | 51.1 | 2,489 | 7.0 | 33,039 | 93.0 | |
| Low | 8,939 | 12.0 | 567 | 6.3 | 8,372 | 93.7 | | 8,371 | 12.0 | 764 | 9.1 | 7,606 | 90.9 | |
| **Secondhand smoke experience** | | | | | | | < .0001 | | | | | | | < .0001 |
| Yes | 38,635 | 51.7 | 2,305 | 6.0 | 36,330 | 94.0 | | 44,916 | 64.6 | 3,583 | 8.0 | 41,333 | 92.0 | |
| No | 36,069 | 48.3 | 1,709 | 4.7 | 34,360 | 95.3 | | 24,563 | 35.4 | 1,509 | 6.1 | 23,054 | 93.9 | |
| **Physical activity** | | | | | | | 0.6851 | | | | | | | 0.0110 |
| No exercise | 19,527 | 26.1 | 1,019 | 5.2 | 18,508 | 94.8 | | 31,117 | 44.8 | 2,205 | 7.1 | 28,912 | 92.9 | |
| 1–3 time(s) in a week | 32,512 | 43.5 | 1,752 | 5.4 | 30,760 | 94.6 | | 29,770 | 42.8 | 2,191 | 7.4 | 27,579 | 92.6 | |
| 4–6 times in a week | 15,722 | 21.0 | 863 | 5.5 | 14,859 | 94.5 | | 6,695 | 9.6 | 533 | 8.0 | 6,162 | 92.0 | |
| Everyday | 6,943 | 9.3 | 380 | 5.5 | 6,563 | 94.5 | | 1,897 | 2.7 | 163 | 8.6 | 1,734 | 91.4 | |
| **Sleeping time** | | | | | | | 0.0595 | | | | | | | < .0001 |
| Sufficient | 54,557 | 73.0 | 2,983 | 5.5 | 51,574 | 94.5 | | 56,479 | 81.3 | 4,254 | 7.5 | 52,225 | 92.5 | |
| Insufficient | 20,147 | 27.0 | 1,031 | 5.1 | 19,116 | 94.9 | | 13,000 | 18.7 | 838 | 6.4 | 12,162 | 93.6 | |
| **Stress** | | | | | | | < .0001 | | | | | | | < .0001 |
| A lot | 22,412 | 30.0 | 1,473 | 6.6 | 20,939 | 93.4 | | 30,800 | 44.3 | 2,624 | 8.5 | 28,176 | 91.5 | |
| A little | 33,103 | 44.3 | 1,729 | 5.2 | 31,374 | 94.8 | | 28,890 | 41.6 | 1,927 | 6.7 | 26,963 | 93.3 | |
| Free | 19,189 | 25.7 | 812 | 4.2 | 18,377 | 95.8 | | 9,789 | 14.1 | 541 | 5.5 | 9,248 | 94.5 | |
| **Subjective health** | | | | | | | < .0001 | | | | | | | < .0001 |
| Healthy | 55,608 | 74.4 | 2,747 | 4.9 | 52,861 | 95.1 | | 44,151 | 63.5 | 2,851 | 6.5 | 41,300 | 93.5 | |
| Normal | 14,553 | 19.5 | 891 | 6.1 | 13,662 | 93.9 | | 19,080 | 27.5 | 1,578 | 8.3 | 17,502 | 91.7 | |
| Unhealthy | 4,543 | 6.1 | 376 | 8.3 | 4,167 | 91.7 | | 6,248 | 9.0 | 663 | 10.6 | 5,585 | 89.4 | |
| **Soda frequency** | | | | | | | 0.1984 | | | | | | | 0.0061 |
| No soda | 12,670 | 17.0 | 680 | 5.4 | 11,990 | 94.6 | | 19,386 | 27.9 | 1,414 | 7.3 | 17,972 | 92.7 | |
| Several in a week | 55,861 | 74.8 | 2,972 | 5.3 | 52,889 | 94.7 | | 47,259 | 68.0 | 3,427 | 7.3 | 43,832 | 92.7 | |
| Several in a day | 6,173 | 8.3 | 362 | 5.9 | 5,811 | 94.1 | | 2,834 | 4.1 | 251 | 8.9 | 2,583 | 91.1 | |
| **Sweet beverage frequency** | | | | | | | 0.0305 | | | | | | | 0.0018 |

(*Continued*)

**Table 1.** (Continued)

| Variables | Atopic dermatitis | | | | | | | | | | | | | |
| --- | --- | --- | --- | --- | --- | --- | --- | --- | --- | --- | --- | --- | --- | --- |
| | Male | | | | | | P-value | Female | | | | | | P-value |
| | Total | | Yes | | No | | | Total | | Yes | | No | | |
| | N | % | N | % | N | % | | N | % | N | % | N | % | |
| No sweet beverage | 9,993 | 13.4 | 488 | 4.9 | 9,505 | 95.1 | | 11,192 | 16.1 | 789 | 7.0 | 10,403 | 93.0 | |
| Several in a week | 56,196 | 75.2 | 3,038 | 5.4 | 53,158 | 94.6 | | 52,423 | 75.5 | 3,807 | 7.3 | 48,616 | 92.7 | |
| Several in a day | 8,515 | 11.4 | 488 | 5.7 | 8,027 | 94.3 | | 5,864 | 8.4 | 496 | 8.5 | 5,368 | 91.5 | |
| **Fast-food frequency** | | | | | | | 0.1776 | | | | | | | 0.2554 |
| No fast-food | 12,759 | 17.1 | 681 | 5.3 | 12,078 | 94.7 | | 13,016 | 18.7 | 977 | 7.5 | 12,039 | 92.5 | |
| Several in a week | 60,490 | 81.0 | 3,239 | 5.4 | 57,251 | 94.6 | | 55,633 | 80.1 | 4,044 | 7.3 | 51,589 | 92.7 | |
| Several in a day | 1,455 | 1.9 | 94 | 6.5 | 1,361 | 93.5 | | 830 | 1.2 | 71 | 8.6 | 759 | 91.4 | |
| **Asthma** | | | | | | | < .0001 | | | | | | | < .0001 |
| Yes | 5,055 | 6.8 | 584 | 11.6 | 4,471 | 88.4 | | 3,775 | 5.4 | 604 | 16.0 | 3,171 | 84.0 | |
| No | 69,649 | 93.2 | 3,430 | 4.9 | 66,219 | 95.1 | | 65,704 | 94.6 | 4,488 | 6.8 | 61,216 | 93.2 | |
| **Rhinitis** | | | | | | | < .0001 | | | | | | | < .0001 |
| Yes | 25,123 | 33.6 | 2,278 | 9.1 | 22,845 | 90.9 | | 24,713 | 35.6 | 2,924 | 11.8 | 21,789 | 88.2 | |
| No | 49,581 | 66.4 | 1,736 | 3.5 | 47,845 | 96.5 | | 44,766 | 64.4 | 2,168 | 4.8 | 42,598 | 95.2 | |
| **Year** | | | | | | | 0.3774 | | | | | | | 0.2923 |
| 2019 | 26,014 | 34.8 | 1,430 | 5.5 | 24,584 | 94.5 | | 24,431 | 35.2 | 1,824 | 7.5 | 22,607 | 92.5 | |
| 2020 | 23,639 | 31.6 | 1,276 | 5.4 | 22,363 | 94.6 | | 21,712 | 31.2 | 1,608 | 7.4 | 20,104 | 92.6 | |
| 2021 | 25,051 | 33.5 | 1,308 | 5.2 | 23,743 | 94.8 | | 23,336 | 33.6 | 1,660 | 7.1 | 21,676 | 92.9 | |

free [female: normal weight (OR: 1.40, CI: 0.96–2.03) and overweight (OR: 1.82, CI: 1.22–2.86)]. Subjective health perception: unhealthy [male: normal weight (OR: 2.04, CI: 1.28–3.23) and overweight (OR: 1.98, CI: 1.26–3.12)]; healthy [female: normal weight (OR: 1.44, CI: 1.19–1.73) and overweight (OR: 1.60, CI: 1.31–1.96)].

In the dietary factors, adolescents with frequency difference in soda, sweet beverage, and fast-food consumption showed variations in the odds of developing AD compared with the underweight groups. Soda frequency: no [male: normal weight (OR: 1.35, CI: 0.92–1.97) and overweight (OR: 1.59, CI: 1.08–2.35); female: normal weight (OR: 1.48 CI: 1.15–1.91) and overweight (OR: 1.57, CI: 1.19–2.07)]. Sweet beverage frequency: several in a day [male: normal weight (OR: 1.62, CI: 1.03–2.54) and overweight (OR: 1.85, CI: 1.14–3.00)]; healthy [female: normal weight (OR: 1.43, CI: 1.01–2.02) and overweight (OR: 1.46, CI: 0.96–2.23)]. Fast-food frequency: several in a week [male: normal weight (OR: 1.19, CI: 1.00–1.41) and overweight (OR: 1.32, CI: 1.10–1.58)]; female: normal weight (OR: 1.24, CI: 1.07–1.44) and overweight (OR: 1.37, CI: 1.16–1.61)].

Fig 1 shows the results of subgroup analysis stratified by diagnosed time of AD. In both male and female adolescents, the normal weight and obese group had higher odds of developing AD compared with the underweight in the order of those who had never been diagnosed with AD, who had been diagnosed with AD but not in the past year [male: normal weight (OR: 1.09, CI: 0.99–1.20), overweight (OR: 1.04, CI: 0.94–1.16); female: normal weight (OR: 1.15, CI: 1.07–1.25), overweight (OR: 1.22, CI: 1.12–1.33)], and who had been diagnosed with AD in the past year [male: normal weight (OR: 1.20, CI: 1.03–1.40), overweight (OR: 1.38, CI: 1.17–1.62); female: normal weight (OR: 1.30, CI: 1.14–1.48), overweight (OR: 1.43, CI: 1.24–1.66)].

Fig 2 shows the results of the subgroup analysis stratified by BMI. Compared with under 5th pct, as the pct section increased (higher BMI), the odds of developing AD increased [male: 5th–25th pct (OR: 1.12, CI: 0.95–1.32), 25th–50th pct (OR: 1.21, CI: 1.03–1.42), 50th–75th pct

**Table 2. Results of factors associated between Body mass index and Atopic dermatitis.**

| Variables | Atopic dermatitis | | | | | | | |
|---|---|---|---|---|---|---|---|---|
| | Male | | | | Female | | | |
| | OR | 95% CI | | | OR | 95% CI | | |
| **Body Mass Index** | | | | | | | | |
| Underweight | 1.00 | | | | 1.00 | | | |
| Normal | 1.19 | (1.02 | - | 1.38) | 1.26 | (1.10 | - | 1.43) |
| Overweight | 1.37 | (1.16 | - | 1.61) | 1.37 | (1.19 | - | 1.58) |
| **Age** | | | | | | | | |
| 7th | 1.00 | | | | 1.00 | | | |
| 8th | 1.05 | (0.92 | - | 1.20) | 1.08 | (0.97 | - | 1.20) |
| 9th | 1.06 | (0.93 | - | 1.21) | 1.06 | (0.94 | - | 1.19) |
| 10th | 1.02 | (0.89 | - | 1.16) | 1.09 | (0.97 | - | 1.21) |
| 11th | 1.06 | (0.93 | - | 1.21) | 1.09 | (0.98 | - | 1.22) |
| 12th | 1.02 | (0.89 | - | 1.17) | 1.06 | (0.94 | - | 1.18) |
| **Region** | | | | | | | | |
| Metropolitan | 1.00 | | | | 1.00 | | | |
| Urban | 1.05 | (0.97 | - | 1.13) | 1.04 | (0.98 | - | 1.11) |
| Rural | 1.10 | (0.96 | - | 1.26) | 1.14 | (0.99 | - | 1.30) |
| **Household income** | | | | | | | | |
| High | 1.00 | | | | 1.00 | | | |
| Middle | 1.03 | (0.96 | - | 1.11) | 0.99 | (0.92 | - | 1.06) |
| Low | 1.20 | (1.07 | - | 1.34) | 1.20 | (1.08 | - | 1.34) |
| **Secondhand smoke experience** | | | | | | | | |
| Yes | 1.18 | (1.10 | - | 1.26) | 1.27 | (1.18 | - | 1.36) |
| No | 1.00 | | | | 1.00 | | | |
| **Physical activity** | | | | | | | | |
| No exercise | 0.89 | (0.78 | - | 1.02) | 0.83 | (0.69 | - | 1.00) |
| 1–3 time(s) in a week | 0.90 | (0.79 | - | 1.01) | 0.88 | (0.74 | - | 1.06) |
| 4–6 times in a week | 0.97 | (0.85 | - | 1.11) | 0.97 | (0.79 | - | 1.20) |
| Everyday | 1.00 | | | | 1.00 | | | |
| **Sleeping time** | | | | | | | | |
| Sufficient | 1.01 | (0.92 | - | 1.11) | 0.96 | (0.87 | - | 1.05) |
| Insufficient | 1.00 | | | | 1.00 | | | |
| **Stress** | | | | | | | | |
| A lot | 1.31 | (1.18 | - | 1.45) | 1.25 | (1.12 | - | 1.40) |
| A little | 1.16 | (1.05 | - | 1.27) | 1.12 | (1.00 | - | 1.26) |
| Free | 1.00 | | | | 1.00 | | | |
| **Subjective health** | | | | | | | | |
| Healthy | 1.00 | | | | 1.00 | | | |
| Normal | 1.15 | (1.05 | - | 1.26) | 1.16 | (1.08 | - | 1.25) |
| Unhealthy | 1.41 | (1.24 | - | 1.61) | 1.33 | (1.20 | - | 1.48) |
| **Soda frequency** | | | | | | | | |
| No soda | 1.00 | | | | 1.00 | | | |
| Several in a week | 0.99 | (0.89 | - | 1.09) | 1.00 | (0.93 | - | 1.08) |
| Several in a day | 1.09 | (0.93 | - | 1.28) | 1.14 | (0.97 | - | 1.34) |
| **Sweet beverage frequency** | | | | | | | | |
| No sweet beverage | 1.00 | | | | 1.00 | | | |
| Several in a week | 1.14 | (1.02 | - | 1.27) | 1.00 | (0.92 | - | 1.10) |

(*Continued*)

**Table 2.** (Continued)

| Variables | Atopic dermatitis | | | | | | | |
|---|---|---|---|---|---|---|---|---|
| | Male | | | | Female | | | |
| | OR | 95% CI | | | OR | 95% CI | | |
| Several in a day | 1.13 | (0.97 | - | 1.31) | 1.05 | (0.92 | - | 1.20) |
| **Fast-food frequency** | | | | | | | | |
| No fast-food | 1.00 | | | | 1.00 | | | |
| Several in a week | 0.97 | (0.88 | - | 1.06) | 0.95 | (0.87 | - | 1.03) |
| Several in a day | 1.04 | (0.80 | - | 1.34) | 0.90 | (0.68 | - | 1.20) |
| **Asthma** | | | | | | | | |
| Yes | 1.83 | (1.65 | - | 2.03) | 1.86 | (1.68 | - | 2.06) |
| No | 1.00 | | | | 1.00 | | | |
| **Rhinitis** | | | | | | | | |
| Yes | 2.49 | (2.33 | - | 2.67) | 2.45 | (2.30 | - | 2.61) |
| No | 1.00 | | | | 1.00 | | | |
| **Year** | | | | | | | | |
| 2019 | 1.02 | (0.93 | - | 1.11) | 1.01 | (0.93 | - | 1.09) |
| 2020 | 1.03 | (0.95 | - | 1.13) | 1.08 | (1.00 | - | 1.16) |
| 2021 | 1.00 | | | | 1.00 | | | |

(OR: 1.22, CI: 1.03–1.44), 75th–95th pct (OR: 1.33, CI: 1.13–1.57), and over 95th pct (OR: 1.34, CI: 1.13–1.60); female: 5th–25th pct (OR: 1.22, CI: 1.06–1.40), 25th–50th pct (OR: 1.23, CI: 1.07–1.42), 50th–75th pct (OR: 1.34, CI: 1.16–1.55), 75th–95th pct (OR: 1.32, CI: 1.14–1.54), and over 95th pct (OR: 1.34, CI: 1.14–1.57)].

## Discussion

We found that the likelihood of developing AD were higher for adolescents with a higher BMI. The mechanisms underlying the association between obesity and AD could be explained by several hypotheses. Obesity could be linked to skin barrier dysfunction, which triggers the occurrence of chronic inflammation and thus leads to the development of AD. Heredity, epidermal dysfunction, skin microbiome abnormalities, and T-cell-induced inflammation (type 2 skewed immune dysregulation is dominant) are involved in the complicated AD pathophysiology [18, 29].

Previous studies with conflicting findings could be attributed to variations in study designs, such as differences in sample sizes and measurement tools for AD and obesity [30]. Furthermore, considering the substantial variation in the prevalence of AD in relation to the age of infants and adolescents [31], the disparity in age among the participants in the study could also be a significant factor contributing to the incongruous results. Meanwhile, prior works have indicated that individuals who are underweight are less likely to develop AD compared to those of normal weight. Nevertheless, these studies have not offered a specific interpretation for this association [19–21].

One Ghanaian study that consistent with the findings of the current study expounds upon the notion that individuals who are underweight exhibit a lower prevalence of AD compared to those with normal weight [32]. This phenomenon can be attributed to the intricate mechanisms involving the reduction of inflammatory markers from underweight status, thereby bolstering the immune system's resistance to foreign substances [32]. In a Japanese study, the fact that the prevalence of wheezing and asthma exhibited a U-shaped pattern, indicating a greater prevalence in individuals who are underweight or overweight than those of normal weight, but

**Table 3. Results of subgroup analysis stratified by independent variables.**

| Variables | Atopic dermatitis | | | | | | | | | |
|---|---|---|---|---|---|---|---|---|---|---|
| | Male | | | | | Female | | | | |
| | Underweight | Normal | | Overweight | | Underweight | Normal | | Overweight | |
| | OR | OR | 95% CI | OR | 95% CI | OR | OR | 95% CI | OR | 95% CI |
| **Physical activity** | | | | | | | | | | |
| No exercise | 1.00 | 1.55 | (1.16 - 2.07) | 1.62 | (1.18 - 2.21) | 1.00 | 1.23 | (1.02 - 1.47) | 1.45 | (1.19 - 1.77) |
| 1–3 time(s) in a week | 1.00 | 1.08 | (0.87 - 1.34) | 1.24 | (0.99 - 1.55) | 1.00 | 1.30 | (1.07 - 1.59) | 1.32 | (1.06 - 1.64) |
| 4–6 times in a week | 1.00 | 0.93 | (0.65 - 1.34) | 1.13 | (0.78 - 1.65) | 1.00 | 1.42 | (0.90 - 2.24) | 1.50 | (0.92 - 2.45) |
| Everyday | 1.00 | 1.48 | (0.75 - 2.92) | 2.03 | (1.02 - 4.05) | 1.00 | 0.86 | (0.47 - 1.54) | 1.10 | (0.55 - 2.20) |
| **Sleeping time** | | | | | | | | | | |
| Sufficient | 1.00 | 1.07 | (0.80 - 1.43) | 1.34 | (0.99 - 1.82) | 1.00 | 1.09 | (0.84 - 1.43) | 0.97 | (0.70 - 1.34) |
| Insufficient | 1.00 | 1.23 | (1.03 - 1.47) | 1.38 | (1.14 - 1.67) | 1.00 | 1.29 | (1.12 - 1.49) | 1.46 | (1.24 - 1.71) |
| **Stress** | | | | | | | | | | |
| A lot | 1.00 | 1.55 | (1.18 - 2.03) | 1.62 | (1.23 - 2.14) | 1.00 | 1.13 | (0.95 - 1.36) | 1.24 | (1.03 - 1.49) |
| A little | 1.00 | 1.03 | (0.82 - 1.29) | 1.21 | (0.95 - 1.54) | 1.00 | 1.39 | (1.12 - 1.71) | 1.44 | (1.14 - 1.81) |
| Free | 1.00 | 1.07 | (0.78 - 1.46) | 1.38 | (0.99 - 1.92) | 1.00 | 1.40 | (0.96 - 2.03) | 1.86 | (1.22 - 2.86) |
| **Subjective health** | | | | | | | | | | |
| Healthy | 1.00 | 1.15 | (0.94 - 1.39) | 1.32 | (1.07 - 1.63) | 1.00 | 1.44 | (1.19 - 1.73) | 1.60 | (1.31 - 1.96) |
| Normal | 1.00 | 1.04 | (0.79 - 1.36) | 1.28 | (0.96 - 1.71) | 1.00 | 1.10 | (0.88 - 1.37) | 1.27 | (1.00 - 1.61) |
| Unhealthy | 1.00 | 2.04 | (1.28 - 3.23) | 1.98 | (1.26 - 3.12) | 1.00 | 1.10 | (0.81 - 1.49) | 1.00 | (0.70 - 1.43) |
| **Soda frequency** | | | | | | | | | | |
| No soda | 1.00 | 1.35 | (0.92 - 1.97) | 1.59 | (1.08 - 2.35) | 1.00 | 1.48 | (1.15 - 1.91) | 1.57 | (1.19 - 2.07) |
| Several in a week | 1.00 | 1.12 | (0.94 - 1.33) | 1.31 | (1.09 - 1.58) | 1.00 | 1.18 | (1.00 - 1.38) | 1.32 | (1.10 - 1.57) |
| Several in a day | 1.00 | 1.48 | (0.90 - 2.45) | 1.41 | (0.82 - 2.41) | 1.00 | 1.35 | (0.86 - 2.12) | 1.14 | (0.64 - 2.06) |
| **Sweet beverage frequency** | | | | | | | | | | |
| No sweet beverage | 1.00 | 1.19 | (0.79 - 1.81) | 1.25 | (0.81 - 1.95) | 1.00 | 1.20 | (0.86 - 1.66) | 1.19 | (0.82 - 1.72) |
| Several in a week | 1.00 | 1.13 | (0.95 - 1.35) | 1.33 | (1.10 - 1.60) | 1.00 | 1.24 | (1.06 - 1.44) | 1.39 | (1.17 - 1.64) |
| Several in a day | 1.00 | 1.62 | (1.03 - 2.54) | 1.85 | (1.14 - 3.00) | 1.00 | 1.43 | (1.01 - 2.02) | 1.46 | (0.96 - 2.23) |
| **Fast-food frequency** | | | | | | | | | | |
| No fast-food | 1.00 | 1.19 | (0.82 - 1.73) | 1.58 | (1.08 - 2.31) | 1.00 | 1.29 | (0.97 - 1.71) | 1.38 | (1.01 - 1.89) |
| Several in a week | 1.00 | 1.19 | (1.00 - 1.41) | 1.32 | (1.10 - 1.58) | 1.00 | 1.24 | (1.07 - 1.44) | 1.37 | (1.16 - 1.61) |
| Several in a day | 1.00 | 1.09 | (0.44 - 2.70) | 1.54 | (0.59 - 4.02) | 1.00 | 1.83 | (0.68 - 4.92) | 1.12 | (0.34 - 3.72) |

not in AD may uphold the linear correlation within this research [33]. These studies also showed slight differences when compared to the present study, as they are contingent upon several factors such as sample size, the range of ages included in the study population, and the specific measurement tools utilized to gather data and information.

In another potential explanation, according to a study conducted on mice, Sirtuin 1 (SIRT1) is essential for the maintenance of the skin barrier [34]. A separate investigation documented that the expression of SIRT1 exhibited a decline in individuals affected by AD, thereby implying a decrease in the capacity of the skin to regenerate [35]. Additionally, higher BMI was linked to lower expression of SIRT1 [36]; therefore, the pattern of SIRT1 expression is inversely associated with the percentage of fat mass. Regarding skin regeneration, a study found that there is an association between elevated BMI and a lower expression of SIRT1 [36], which is linked to AD. This suggests that obese adolescents, as well as those with normal BMI who are diagnosed with AD may benefit from weight management. BMI, which is known to be fluctuate during adolescence [37], may have influenced the outcome of recent AD diagnosis. In individuals diagnosed with AD throughout their lives but not within the last year, there may have been a difference in

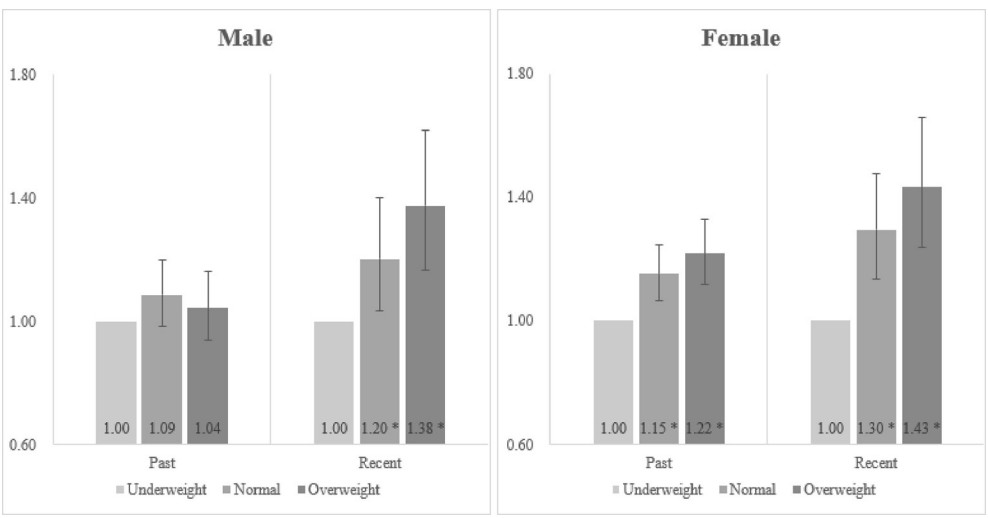

**Fig 1. Results of subgroup analysis stratified by diagnosed time of atopic dermatitis.** Error bars: 95% confidence interval. *: p-value < .05 Reference: underweight Past: who have been diagnosed with atopic dermatitis but not in the past 1 year. Recent: who have been diagnosed with atopic dermatitis in the past 1-year.

BMI between past and current assessments. Additionally, individuals with obesity may have sought medical attention in the past year due to the severity of AD [38].

In the other factors that contribute to the association between the degree of obesity and AD, individuals with lesser levels of physical activity and higher levels of BMI had increased likelihoods of developing AD. Patients with AD may refrained from performing exercise or vigorous physical activity due to the fact that intense physical activity can produce sweat and provoke itchiness [13]. Still, the symptoms of AD can be mitigated through exercise, making it a viable approach for managing weight [39]. Constant AD symptoms can disrupt in sleeping patterns, resulting in insufficient duration of sleep [40]. Additionally, inadequate sleep has been associated with obesity, exacerbating the severity of AD [41].

Gender modulating effects were also observed between these associations. With high BMI, elevated levels of subjective stress in males whereas lower levels of that in females were

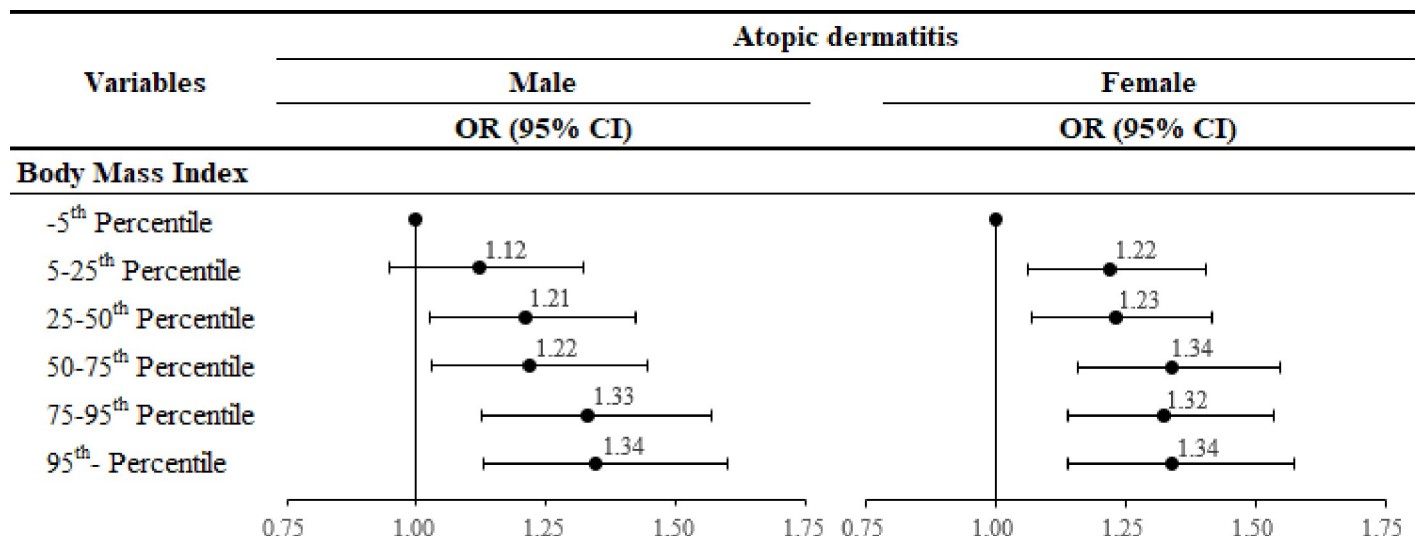

**Fig 2. Results of subgroup analysis stratified by body mass index.** OR: odds ratio. CI: confidence interval.

associated with greater likelihoods of AD. Research on stress levels and the atopic risk with similar results of the present study proposed young men exhibit more prominent cortisol responses to stress than young women [42]. Furthermore, subjective health perception may be a connected factor to the findings of the present study. Specifically, males in unhealthy groups demonstrated a higher likelihood of AD, whereas women in healthy groups exhibited a similar trend, highlighting the impact of stress on AD symptoms. Nonetheless, further investigation is required to better understand this association, particularly in terms of gender disparities. As for dietary factors, the implementation of a strategy to restrict the consumption of certain foods that have been indicated to exacerbate symptoms of AD as a preventive measure may associated with reduced intake of soft drinks [43]. Nevertheless, an increased frequency of consumption of sweet beverage or fast-food is associated with a heightened likelihood of developing AD, implying that such options may be considered restrictive in terms of weight management [44, 45].

This study has the strength of being capable of generalizing findings to South Korean adolescents by utilizing nationally representative KYRBS data. Furthermore, the multiple analyses conducted in this study is notable for examining the correlation with underweight, an issue that has received less attention in previous studies. However, this study has several limitations. First, this was a cross-sectional study and may be susceptible to reverse causality. Bidirectional situations can be possible; that is, having a higher BMI may predispose an individual to develop AD, and AD may cause an elevated BMI [46]. Second, the questionnaire used in this study conducted by KYRBS was a self-reported questionnaire. Thereby, the participants were likely to provide incorrect answers either intentionally or unintentionally. Third, due to the availability of surveys, adjusting for potential confounders, such as the severity of AD, other dietary compositions associated with AD, living environment, or allergic history may not be sufficient.

In conclusion, this study was conducted to examine the association between the degree of obesity measured by BMI and AD in Korean adolescents. The normal-weight and obese adolescents had higher likelihood of developing AD compared with the underweight adolescents both in males and females. These results may provide information about the feasibility of weight reduction as a non-pharmacological treatment for AD.

## Author Contributions

**Conceptualization:** Jae Hyeok Lim, Yun Seo Jang, Eun-Cheol Park.

**Data curation:** Jae Hyeok Lim.

**Formal analysis:** Jae Hyeok Lim, Yun Seo Jang, Suk-Yong Jang, Eun-Cheol Park.

**Investigation:** Jae Hyeok Lim, Suk-Yong Jang.

**Methodology:** Jae Hyeok Lim, Yun Seo Jang, Dan Bi Kim, Suk-Yong Jang, Eun-Cheol Park.

**Project administration:** Dan Bi Kim.

**Supervision:** Eun-Cheol Park.

**Validation:** Jae Hyeok Lim.

**Visualization:** Jae Hyeok Lim, Eun-Cheol Park.

**Writing – original draft:** Jae Hyeok Lim.

**Writing – review & editing:** Jae Hyeok Lim.

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
