## [Decision Letter · Decision Letter 0]

11 Apr 2024

PONE-D-24-07867Association between degree of obesity and atopic dermatitis among adolescents: Findings from a national cross-sectional study in KoreaPLOS ONE

Dear Dr. Park,

Thank you for submitting your manuscript to PLOS ONE. After careful consideration, we feel that it has merit but does not fully meet PLOS ONE’s publication criteria as it currently stands. Therefore, we invite you to submit a revised version of the manuscript that addresses the points raised during the review process.

We look forward to receiving your revised manuscript.

Kind regards,

Dong Keon Yon, MD, FACAAI, FAAAAI

Academic Editor

PLOS ONE

Journal Requirements:

"This study did not receive any specific grant from funding agencies in the public, commercial, or not-for-profit sectors."

Additional Editor Comment:

Thank you for submitting your manuscript. The reviewers and I believe it is of potential value for our readers. However, the reviewers have raised a number of very important issues, and their excellent comments will need to be adequately addressed in a revision before the acceptability of your manuscript for publication in the Journal can be determined. We cannot guarantee that your revised paper will be chosen for publication; this would be solely based on how satisfactorily you have addressed the reviewer comments.

#1. KCDC -> KDCA

#2. To determine the association between BMI and AD, multiple logistic regression analysis was performed after adjusting for all covariates. -> Please cite the statistical guideline (DOI: https://doi.org/10.54724/lc.2022.e3).

Reviewers' comments:

Reviewer's Responses to Questions

**Comments to the Author**

1. Is the manuscript technically sound, and do the data support the conclusions?

Reviewer #1: Partly

Reviewer #2: Yes

2. Has the statistical analysis been performed appropriately and rigorously? 

Reviewer #1: Yes

Reviewer #2: I Don't Know

3. Have the authors made all data underlying the findings in their manuscript fully available?

Reviewer #1: Yes

Reviewer #2: Yes

4. Is the manuscript presented in an intelligible fashion and written in standard English?

Reviewer #1: Yes

Reviewer #2: Yes

5. Review Comments to the Author

Reviewer #1: This is a good manuscript. But I have some comments as followed.

1.The title might be revise as "Association between BMI and atopic dermatitis among adolescents: Findings from a national cross-sectional study in Korea".

2.Background part: I suggest describe the epidemical situation of atopic dermatitis at first.

3.Methods: part: Please describe the measure methods of the important variables. How to measure obesity and underweight?

4.Please add the inclusion criteria and exclusion criteria in the part of Participants.

5.Results part: Please add to the subtitle of different part.

6.Discussion part: please add to advantage of the research before the limitation of the research.

Reviewer #2: Through this manuscript, the authors have effectively demonstrated the relationship between the degree of obesity and atopic dermatitis. I appreciated how they tied this association to the clinical aspect of the disease, offering non-pharmacological interventions as a viable option. However, upon reviewing the manuscript, I encountered some conflicting points. I hope the authors will view this feedback positively.

1. In the Materials and Methods section, under the subsection "Data," the authors mentioned "sampling design method." I intend to question the appropriateness of this sampling technique.

2. Under the subheading "Variables," the study mentioned using BMI levels categorized as underweight, normal weight, and obesity. However, the World Health Organization classifies BMI into underweight, normal, overweight, and obese categories. Therefore, it would enhance the study's credibility to align with the globally accepted classification system.

3. Atopic dermatitis is a chronic skin condition. However, in this study, participants with a past history of atopic dermatitis (more than 1 year ago) were categorized into the AD "no" group. I believe this could potentially underestimate the prevalence of the disease, which might impact the subsequent analyses conducted in the study.

4.In the majority of cases, atopic dermatitis (AD) has an onset before the age of five years. However, this study targeted participants aged 12 to 18 years. This age group, coupled with the exclusion of participants with a history of AD, could have influenced the results, particularly in terms of prevalence.

5.The authors stated that "All participants signed an informed consent when they participated." However, it's worth noting that the majority of participants are minors. This raises ethical concerns regarding whether it is appropriate for a study to involve minors signing consent forms.

6.The data provided in the manuscript are publicly available on the website mentioned. However, it's important to note that all the data are in a language other than English. If feasible, it would be beneficial to have the data translated into English for wider accessibility and comprehension.

6. PLOS authors have the option to publish the peer review history of their article (what does this mean?). If published, this will include your full peer review and any attached files.

Reviewer #1: No

Reviewer #2: No

---

## [Author Response · Author response to Decision Letter 0]

22 Jun 2024

We express our gratitude for the opportunity to make revisions to our manuscript. Throughout the revision process, we have taken into careful consideration the feedback provided by the editor and reviewers, and have made every effort to integrate their suggestions appropriately. Following the given instructions, we have endeavored to elucidate the modifications implemented in response to all reviewer comments. The insights offered by the reviewers were beneficial, and we value the constructive criticism provided for our submission. Upon addressing the raised concerns, we believe that the quality of the paper has significantly improved, and we trust that you share this sentiment. Our detailed responses to each comment are outlined below, and we have enclosed a revised version of the manuscript with the tracked changes. Once again, we appreciate the insightful and constructive feedback.

---

## [Editor Report · Decision Letter 1]

2 Jul 2024

Association between degree of obesity and atopic dermatitis among adolescents: Findings from a national cross-sectional study in Korea

PONE-D-24-07867R1

Dear Dr. Park,

We’re pleased to inform you that your manuscript has been judged scientifically suitable for publication and will be formally accepted for publication once it meets all outstanding technical requirements.

Kind regards,

Dong Keon Yon, MD, FACAAI, FAAAAI

Academic Editor

PLOS ONE

Additional Editor Comments (optional):

This is an excellent paper.
---

## [Editor Report · Acceptance letter]

9 Jul 2024

PONE-D-24-07867R1 

PLOS ONE

Dear Dr. Park, 

I'm pleased to inform you that your manuscript has been deemed suitable for publication in PLOS ONE. Congratulations! Your manuscript is now being handed over to our production team.

Kind regards, 

on behalf of

Dr. Dong Keon Yon 

Academic Editor

PLOS ONE